# EquiMod: An Equivariance Module to Improve Visual Instance Discrimination

**Alexandre Devillers & Mathieu Lefort**
Univ Lyon, UCBL, CNRS, INSA Lyon
LIRIS, UMR5205, F-69622
Villeurbanne, France
{alexandre.devillers,mathieu.lefort}@liris.cnrs.fr

## Abstract

Recent self-supervised visual representation methods are closing the gap with su-pervised learning performance. Most of these successful methods rely on max-imizing the similarity between embeddings of related synthetic inputs created through data augmentations. This can be seen as a task that encourages embed-dings to leave out factors modified by these augmentations, i.e. to be invariant to them. However, this only considers one side of the trade-off in the choice of the augmentations: they need to strongly modify the images to avoid simple so-lution shortcut learning (e.g. using only color histograms), but on the other hand, augmentations-related information may be lacking in the representations for some downstream tasks (e.g. literature shows that color is important for bird and flower classification). Few recent works proposed to mitigate this problem of using only an invariance task by exploring some form of equivariance to augmentations. This has been performed by learning additional embeddings space(s), where some aug-mentation(s) cause embeddings to differ, yet in a non-controlled way. In this work, we introduce *EquiMod* a generic equivariance module that structures the learned latent space, in the sense that our module learns to predict the displacement in the embedding space caused by the augmentations. We show that applying that mod-ule to state-of-the-art invariance models, such as BYOL and SimCLR, increases the performances on the usual CIFAR10 and ImageNet datasets. Moreover, while our model could collapse to a trivial equivariance, i.e. invariance, we observe that it instead automatically learns to keep some augmentations-related information beneficial to the representations.

Source code is available at https://github.com/ADevillers/EquiMod

## 1 Introduction

Using relevant and general representation is central for achieving good performances on downstream tasks, for instance when learning object recognition from high-dimensional data like images. His-torically, feature engineering was the usual way of building representations, but we can currently rely on deep learning solutions to automate and improve this process of representation learning. Still, it is challenging as it requires learning a structured latent space while controlling the precise amount of features to put in representations: too little information will lead to not interesting rep-resentations, yet too many non-pertinent features will make it harder for the model to generalize. Recent works have focused on Self-Supervised Learning (SSL), i.e. determining a supervisory sig-nal from the data with a pretext task. It has the advantages of not biasing the learned representation toward a downstream goal, as well as not requiring human labeling, allowing the use of plentiful raw data, especially for domains lacking annotations. In addition, deep representation learning en-courages network reuse via transfer learning, allowing for better data efficiency and lowering the computational cost of training for downstream tasks compared to the usual end-to-end fashion.

The performances of recent instance discrimination approaches in SSL of visual representation are progressively closing the gap with the supervised baseline (Caron et al., 2020; Chen et al., 2020a;b;

Chen & He, 2021; Bardes et al., 2021; Grill et al., 2020; He et al., 2020; Misra & Maaten, 2020; Zbontar et al., 2021). They are mainly siamese networks performing an instance discrimination task. Still, they have various distinctions that make them different from each other (see Liu (2021) for a review and Szegedy et al. (2013) for a unification of existing works). Their underlying mechanism is to maximize the similarity between the embedding of related synthetic inputs, a.k.a. views, created through data augmentations that share the same concepts while using various tricks to avoid a collapse towards a constant solution (Jing et al., 2021; Hua et al., 2021). This induces that the latent space learns an invariance to the transformations used, which causes representations to lack augmentations-related information.

Even if these models are self-supervised, they rely on human expert knowledge to find these relevant invariances. For instance, as most downstream tasks in computer vision require object recognition, existing augmentations do not degrade the categories of objects in images. More precisely, the choice of the transformations was driven by some form of supervision, as it was done by experimentally searching for the set of augmentations giving the highest object recognition performance on the ImageNet dataset (Chen et al., 2020a). For instance, it has been found that color jitter is the most efficient augmentation on ImageNet. One possible explanation is that color histograms are an easy-to-learn shortcut solution (Geirhos et al., 2020), which is not removed by cropping augmentations (Chen et al., 2020a). Indeed, as there are many objects in the categories of ImageNet, and as an object category does not change when its color does, the loss of color information is worth removing the shortcut. Still, it has been shown that color is an essential feature for some downstream tasks (Xiao et al., 2020).

Thus, for a given downstream task, we can separate augmentations into two groups: the ones for which the representations benefit from insensitivity (or invariance) and the ones for which sensitivity (or variance) is beneficial (Dangovski et al., 2021). Indeed, there is a trade-off in the choice of the augmentations: they require to modify significantly the images to avoid simple solution shortcut learning (e.g. relying just on color histograms), yet some downstream tasks may need augmentations-related information in the representations. Theoretically, this trade-off limits the generalization of such representation learning methods relying on invariance. Recently, some works have explored different ways of including sensitivity to augmentations and successfully improved augmentations-invariant SSL methods on object classification by using tasks forcing sensitivity while keeping an invariance objective in parallel. Dangovski et al. (2021) impose a sensitivity to rotations, an augmentation that is not beneficial for the invariance task, while we focus in this paper on sensitivity to transformations used for invariance. Xiao et al. (2020) proposes to learn as many tasks as there are augmentations by learning multiple latent spaces, each one being invariant to all but one transformation, however, it does not control the way augmentations-related information is conserved. One can see this as an implicit way of learning variance to each possible augmentation. Contrary to these works that do not control the way augmentations-related information is conserved, here we propose to explore sensitivity by introducing an equivariance module that structures its latent space by learning to predict the displacement in the embedding space caused by augmentations in the pixel space.

The contributions of this article are the following:

- We introduce a generic equivariance module *EquiMod* to mitigate the invariance to augmentations in recent methods of visual instance discrimination;

- We show that using *EquiMod* with state-of-the-art invariance models, such as BYOL and SimCLR, boosts the classification performances on CIFAR10 and ImageNet datasets;

- We study the robustness of *EquiMod* to architectural variations of its sub-components;

- We observe that our model automatically learns a specific level of equivariance for each augmentation.

Sec. 2 will present our EquiMod module as well as the implementation details while in Sec. 3 we will describe the experimental setup used to study our model and present the results obtained. The Sec. 4 will position our work w.r.t. related work. Finally, in Sec. 5 we will discuss our current results and possible future works.

## 2 EQUIMOD

### 2.1 NOTIONS OF INVARIANCE AND EQUIVARIANCE

As in Dangovski et al. (2021), we relate the notions of augmentations sensitivity and insensitivity to the mathematical concepts of invariance and equivariance. Let $\mathcal{T}$ be a distribution of possible transformations, and $f$ denotes a projection from the input space to a latent space. That latent space is said to be invariant to $\mathcal{T}$ if for any given input $\boldsymbol{x}$ the Eq. 1 is respected.

$$\forall t \in \mathcal{T} \qquad f(t(\boldsymbol{x})) = f(\boldsymbol{x}) \tag{1}$$

Misra & Maaten (2020) used this definition of invariance to design a pretext task for representation learning. This formulation reflects that the embedding of a non-augmented input sample $\boldsymbol{x}$ will not change if the input is transformed by any of the transformations in $\mathcal{T}$. However, more recent works (Bardes et al., 2021; Chen et al., 2020a; Chen & He, 2021; Grill et al., 2020; Zbontar et al., 2021) focused on another formulation of invariance defined by the following Eq. 2.

$$\forall t \in \mathcal{T}, \ \forall t' \in \mathcal{T} \qquad f(t(\boldsymbol{x})) = f(t'(\boldsymbol{x})) \tag{2}$$

With this definition, the embedding produced by an augmented sample $\boldsymbol{x}$ is independent of the transformation used. Still, note that Eq. 1 implies Eq. 2, and that if the identity function is part of $\mathcal{T}$, which is the case with recent approaches, then both definitions are indeed equivalent.

While insensitivity to augmentation is reflected by invariance, sensitivity can be obtained by achieving variance, i.e. replacing the equality by inequality in Eq. 1 or Eq. 2. Yet, this is not an interesting property, as any injective function will satisfy this constraint. In this paper, we propose to use equivariance as a way to achieve variance to augmentations for structuring our latent space. Eq. 3 gives the definition of equivariance used in the following work.

$$\forall t \in \mathcal{T}, \ \exists u_t \qquad f(t(\boldsymbol{x})) = u_t(f(\boldsymbol{x})) \tag{3}$$

With $u_t$ being a transformation in the latent space parameterized by the transformation $t$, it can be seen as the counterpart of the transformation $t$ but in the embedding space. With this definition, the embeddings from different augmentations will be different and thus encode somehow information related to the augmentations. Yet, if $u_t$ is always the identity then this definition of equivariance becomes the same as invariance Eq.1. Indeed, one can see invariance as a trivial specific case of equivariance. In the following, we only target non-trivial equivariance where $u_t$ produces some displacement in the latent space. See Fig. 1 for a visual comparison of invariance and equivariance.

### 2.2 METHOD

*EquiMod* is a generic equivariance module that acts as a complement to existing visual instance discrimination methods performing invariance (Bardes et al., 2021; Chen et al., 2020a; Chen & He, 2021; Grill et al., 2020; Zbontar et al., 2021). The objective of this module is to capture some augmentations-related information originally suppressed by the learned invariance to improve the learned representation. The main idea relies on equivariance, in the sense that our module learns to predict the displacement in the embedding space caused by the augmentations. This way, by having non-null displacement, we ensure embeddings contain augmentations-related information. We first introduce a formalization for these existing methods (see Bardes et al. (2021) for an in-depth explanation of this unification), before introducing how our approach adds on top.

Let $t$ and $t'$ denote two augmentations sampled from the augmentations distribution $\mathcal{T}$. For the given input image $\boldsymbol{x}$, two views are defined as $\boldsymbol{v}_i \coloneqq t(\boldsymbol{x})$ and $\boldsymbol{v}_j \coloneqq t'(\boldsymbol{x})$. Thus, for $N$ original images, this results in a batch of $2N$ views, where the first $N$ elements correspond to a first view ($\boldsymbol{v}_i$) for each of the images, and the last $N$ elements correspond to a second view ($\boldsymbol{v}_j$) for each of the images. Following previous works, we note $f_\theta$ an encoder parameterized by $\theta$ producing representations from images, and $g_\phi$ a projection head parameterized by $\phi$, which projects representations in an embedding space. This way, the representations are defined as $\boldsymbol{h}_i \coloneqq f_\theta(\boldsymbol{v}_i)$ as well as $\boldsymbol{h}_j \coloneqq f_\theta(\boldsymbol{v}_j)$,

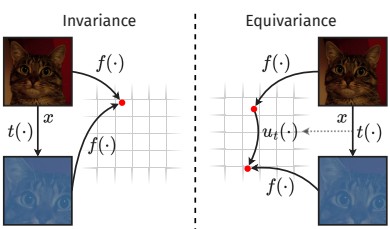

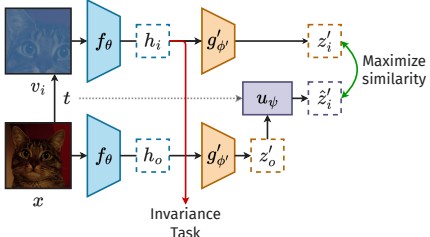

Figure 2: The model learns similar embeddings for an augmented view ($z_i'$) and the prediction of the displacement in the embedding space caused by that augmentation ($\hat{z}_i'$), $t$ is a learned representation of the parameters of the transformation, see Sec. 2 for notation details.

Figure 1: On the left, invariance described by Eq. 1, on the right, equivariance considered in this paper and described by Eq. 3.

and the embeddings as $z_i := g_\phi(h_i)$ as well as $z_j := g_\phi(h_j)$. Then, the model learns to maximize the similarity between $z_i$ and $z_j$, while using diverse tricks to maintain a high entropy for the embeddings, preventing collapse to constant representations.

To extend those preceding works, we introduce a second latent space to learn our equivariance task. For this purpose, we first define a second projection head $g_{\phi'}'$ parameterized by $\phi'$ whose objective is to project representations in our latent space. Using this projection head we note $z_i' := g_{\phi'}'(h_i)$ and $z_j' := g_{\phi'}'(h_j)$, the embeddings of the views $v_i$ and $v_j$ in this latent space we introduce. Moreover, the way we define equivariance in Eq 3 requires us to produce the embedding of the non-augmented image $x$, thus we note the representation $h_o := f_\theta(x)$, which is used to create the embedding $z_o' := g_{\phi'}'(h_o)$ for the given image $x$. Next, as mentioned in Sec.2.1, to learn an equivariant latent space one needs to determine a transformation $u_t$ for any given $t$, this can be done either by fixing it or by learning it. In this work, we learn the transformation $u_t$. To this end, we define $u_\psi$ a projection parameterized by the learnable parameters $\psi$, referenced later as the equivariance predictor (implementation details about how $t$ is encoded and influences $u_\psi$ are given below in Sec. 2.3). The goal of this predictor is to produce $\hat{z}_i'$ from a given $z_o'$ and $t$ (resp. $\hat{z}_j'$ for $z_o'$ and $t'$). One can see $\hat{z}_i'$ as an alternative way to obtain $z_i'$ using the equivariance property defined by Eq. 3. Instead of computing the embedding of the augmented view $v_i := t(x)$, we apply $t$ via $u_\psi$ on the embedding $z_o'$ of the original image $x$.

Therefore, to match this equivariance principle, we need to train $g_{\phi'}'$ and $u_\psi$ so that applying the transformation via a predictor in the latent space ($\hat{z}_i'$) is similar to applying the transformation in the input space and then computing the embedding ($z_i'$). For this purpose, we denote ($z_i', \hat{z}_i'$) as positive pair (resp. ($z_j', \hat{z}_j'$)), and design our equivariance task so that our model learns to maximize the similarity between the positive pairs. Yet, one issue with this formulation is that it allows collapsed solutions, e.g. every $z'$ being a constant. To avoid such simple solutions, we consider negative pairs (as in Chen et al. (2020a); He et al. (2020)) to repulse embedding from other embedding coming from views of different images. We use the Normalized Temperature-scaled cross entropy (NT-Xent) loss to learn from these positive and negative pairs, thus defining our equivariance loss for the positive pair of the invariance loss $(i,j)$ as Eq. 4:

$$\ell_{i,j}^{EquiMod} = -\log \frac{\exp(\text{sim}(z_i', \hat{z}_i')/\tau')}{\sum_{k=1}^{2N} \mathbb{1}_{[k \neq i \wedge k \neq j]} \exp(\text{sim}(z_i', z_k')/\tau')} \tag{4}$$

where $\tau'$ is a temperature parameter, $\text{sim}(a, b)$ is the cosine similarity defined as $a^\top b/(\|a\|\|b\|)$, and $\mathbb{1}_{[k \neq i \wedge k \neq j]}$ is the indicator function evaluated to 1 (0 otherwise) when $k \neq i$ and $k \neq j$.

This way, we exclude from negative pairs the views of the same image, related to index i and j, that are considered as positive pairs in the invariance methods. While we could consider these pairs as negative and still be following 3, we found that not using them as negative nor as positive leads to slightly better results. One hypothesis is that repulsing views that can be very close in the pixel

space (e.g. if the sampled augmentations modify weakly the original image) could induce training instability. One can notice that $g'_{\phi'}$ and $u_\psi$ are learned simultaneously, thus they can influence each other during the training phase. We finally define the total loss of the model as:

$$\mathcal{L} = \mathcal{L}_{Invariance} + \lambda \mathcal{L}_{EquiMod}$$

with $\mathcal{L}_{EquiMod}$ being the loss Eq. 4 applied to all pairs, both $(i, j)$ and $(j, i)$, of a batch and $\mathcal{L}_{Invariance}$ being the loss of the invariance baseline. $\lambda$ is a hyperparameter that ponders the equivariant term of the loss.

## 2.3 IMPLEMENTATION DETAILS

We tested our module as a complement of 3 different baselines. The first one is SimCLR (Chen et al., 2020a) as it represents a contrastive approach to instance discrimination and performs well on CIFAR. The second one is BYOL (Grill et al., 2020), which offers a different kind of architecture (as it is a bootstrapping approach rather than a contrastive one) while having the highest top-1 accuracy with linear evaluation on ImageNet using a ResNet50 backbone in a self-supervised fashion. We also tested Barlow Twins (Zbontar et al., 2021) as it is not exactly a contrastive approach nor a bootstrapping one to illustrate the generality of our approach, yet limited to CIFAR10 due to computational limitation. Here are the details of each part of the architecture, including the baseline ones and our equivariance module:

- *Encoder*: we follow existing works and use a convolutional neural network for the encoder $f_\theta$, more specifically deep residual architectures from He et al. (2016).

- *Invariance projection head*: for the projection head $g_\phi$ (and potential predictor as in BYOL Grill et al. (2020)), we used the same experimental setups as the original papers, except for SimCLR where we used a 3 layers projection head as in Chen & He (2021).

- *Equivariance projection head*: the setup of our projection head $g'_{\phi'}$ is a 3 layers Multi-Layer Perceptron (MLP), where each Fully-Connected (FC) layer is followed by a Batch Normalization (BN) and a ReLU activation, except the last layer which is only followed by a BN and no ReLU. Hidden layers have 2048 neurons each.

- *Equivariant predictor*: the predictor $u_\psi$ is a FC followed by a BN. Its input is the concatenation of a representation of $t$ and the input embedding $z'_o$. More precisely $t$ is encoded by a numerical learned representation of the parameters that fully define it. More precisely, we reduce the augmentation to a vector composed of binary values related to the use of transformations (for transformations applied with a certain probability) and numerical values corresponding to some parameters (of the parameterized transformations). This vector is projected in a 128d latent space with a perceptron learned jointly with the rest of the model, see Sec.A.1 for details and examples of this encoding. This way, the input dimension of the predictor is the dimension of the latent space plus the dimension of the encoding of augmentations, while the output dimension is the same as the latent space.

## 3 EXPERIMENTS

### 3.1 EXPERIMENTAL SETTINGS

In our experimentations, we tested our method on ImageNet (IN) (Deng et al., 2009) and CIFAR10 (Krizhevsky et al., 2009). As mentioned before, we have used our module as a complement to SimCLR, BYOL, and Barlow Twins, 3 state-of-the-art invariance methods with quite different ideas, to test the genericity of our module. For these methods, we used the same experimental setup as the original papers. As in previous works, while training on ImageNet we used a ResNet50 without the last FC, but while training on CIFAR10 we used the CIFAR variant of ResNet18 (He et al., 2016). For all our experimentations we used the LARS You et al. (2017) optimizer, yet, biases and BN parameters were excluded from both weight decay and LARS adaptation as in Grill et al. (2020)). Finally, we have fixed $\lambda$ to 1 as it led to the best and more stable results.

### 3.1.1 SIMCLR

The model is trained for 800 epochs with 10 warm-up epochs and a cosine decay learning rate schedule. We have used a batch size of 4096 for ImageNet and 512 for CIFAR10, while using an initial learning rate of 2.4 for ImageNet (where we use 4.8 for SimCLR without EquiMod, as in the original paper) and 4.0 for CIFAR10. For the optimizer, we fix the momentum to 0.9 and the weight decay to $1e^{-6}$. Both the invariant and equivariant latent space dimensions have been set to 128. Finally, we use $\tau' = 0.2$ for our loss, but $\tau = 0.2$ on ImageNet $\tau = 0.5$ with CIFAR10 for the loss of SimCLR (we refer the reader to the original paper for more information about the loss of SimCLR (Chen et al., 2020a)).

### 3.1.2 BYOL

The model learned for 1000 epochs[1] (800 on CIFAR10) with 10 warm-up epochs and a cosine decay learning rate schedule. The batch size used is 4096 for ImageNet and 512 for CIFAR10. We have been using an initial learning rate of 4.8 for ImageNet (where we use 3.2 for BYOL without EquiMod, as in the original paper) while using 2.0 for CIFAR10. Momentum of the optimizer is set to 0.9 and weight decay to $1.5e^{-6}$ on ImageNet, but $1e^{-6}$ on CIFAR10. The invariant space has 256 dimensions while we keep our equivariant latent space to 128. Last, we use $\tau' = 0.2$ for our loss, and $\tau_{\text{base}} = 0.996$ for the momentum encoder of BYOL with a cosine schedule as in the original paper (once again, we refer the reader to the paper for more details (Grill et al., 2020)).

### 3.1.3 BARLOW TWINS

We tested our method with Barlow Twins only on CIFAR10 with the following setup: 800 epochs with 10 warm-up epochs and a cosine decay learning rate schedule, a batch size of 512, an initial learning rate of 1.2, a momentum of 0.9 and weight decay of $1.5e^{-6}$. Both the invariant and equivariant latent space has 128 dimensions, while we use $\tau' = 0.2$ for our loss and $\lambda_{\text{Barlow Twins}} = 0.005$ for the loss of Barlow Twins (as in the original paper (Grill et al., 2020)).

## 3.2 RESULTS

### 3.2.1 LINEAR EVALUATION

After training on either ImageNet or CIFAR10, we evaluate the quality of the learned representation with the linear evaluation which is usual in the literature. To this end, we train a linear classifier on top of the frozen representation, using the Stochastic Gradient Descent (SGD) for 90 epochs, which is sufficient for convergence, with a batch size of 256, a Nesterov momentum of 0.9, no weight decay, an initial learning rate of 0.2 and a cosine decay learning rate schedule.

Results of this linear evaluation are presented in Table 1, while some additional results are present in supplementary material Sec. A.3. Across all baselines and datasets tested, EquiMod increases the performances of all the baselines used, except BYOL while trained on 1000 epochs. Still, it is worth noting that under 100 and 300 epochs training (Sec. A.3), EquiMod improves the performances of BYOL. Overall, this supports the genericity of our approach, and moreover, confirms our idea that adding an equivariance task helps to extract more pertinent information than just an invariance task and improves representations. On CIFAR10, we achieve the second-best performance after E-SSL, yet, contrary to us, they tested their model on an improved hyperparameter setting of SimCLR.

### 3.2.2 EQUIVARIANCE MEASUREMENT

The way our model is formulated could lead to the learning of invariance rather than equivariance. Indeed, learning an invariant latent space as well as the function identity for $u_\psi$ is an admissible solution. Therefore, to verify that our model is really learning equivariance, we define two metrics of equivariance Eq.5 and Eq.6. The first one evaluates the absolute displacement toward $z'_i$ caused by the predictor $u_\psi$. One can see this as how much applying the augmentation $t$ to $z'_o$ in the latent space via $u_\psi$ makes the resulting embedding $\hat{z}'_i$ more similar to $z'_i$. This way, if our model is learning invariance, we should observe an absolute displacement of 0, as $u_\psi$ would be the identity. On the

---

[1]We also performed 100 and 300 epochs training, see Sec. A.3.

| Method | ImageNet | | CIFAR10 | |
|---|---|---|---|---|
| | Top-1 | Top-5 | Top-1 | Top-5 |
| PIRL (Misra & Maaten, 2020) | 63.6 | - | - | - |
| E-SimCLR (Dangovski et al., 2021) | 68.3‡ | - | 94.1 | - |
| E-SimSiam (Dangovski et al., 2021) | 68.6‡ | - | 94.2 | - |
| SimCLR (Chen et al., 2020a) | 69.3 | 89.0 | - | - |
| SimSiam (Chen & He, 2021) | 71.3 | - | - | - |
| SwAV (w/o multi-crop) (Caron et al., 2020) | 71.8 | - | - | - |
| Barlow Twins (Zbontar et al., 2021) | 73.2 | 91.0 | - | - |
| VICReg (Bardes et al., 2021) | 73.2 | 91.1 | - | - |
| BYOL (Grill et al., 2020) | 74.3 | 91.6 | - | - |
| SimCLR∗ | 71.57 | 90.48 | 90.96 | 99.73 |
| SimCLR∗ + EquiMod | **72.30** | **90.84** | **92.79** | **99.78** |
| BYOL∗ | **74.03** | **91.51** | 90.44 | 99.62 |
| BYOL∗ + EquiMod | 73.22 | 91.26 | **91.57** | **99.71** |
| Barlow Twins∗ | - | - | 86.94 | 99.61 |
| Barlow Twins∗ + EquiMod | - | - | **88.87** | **99.71** |

Table 1: **Linear Evaluation**; top-1 and top-5 accuracies (in %) under linear evaluation on ImageNet and CIFAR10 (symbols ∗ denote our re-implementations, and ‡ denote only 100 epochs training).

contrary, if it is learning equivariance, we should observe a positive value, meaning that the $u_\psi$ plays its role in predicting the displacement in the embedding space caused by the augmentations. A negative displacement means a displacement in the opposite direction, in other words, it means that the predictor performs worse than the identity function. Furthermore, a small displacement does not mean poor equivariance, for instance, if $z_o'$ is very similar to $z_i'$, the room for displacement is already very small. This is why we also introduce the second metric, which evaluates the relative displacement toward $z_i'$ caused by $u_\psi$. It reflects by which factor applying the augmentation $t$ to $z_o'$ in the latent space via $u_\psi$ makes the resulting embedding $\hat{z}_i'$ less dissimilar to $z_i'$. Thus, if the model is learning invariance, we should see no reduction nor augmentation of the dissimilarity, thus the factor should remain at 1 while a model achieving equivariance would exhibit a positive factor.

$$\mathrm{sim}(z_i', \hat{z}_i') - \mathrm{sim}(z_i', z_o') \qquad (5) \qquad\qquad \frac{1 - \mathrm{sim}(z_i', z_o')}{1 - \mathrm{sim}(z_i', \hat{z}_i')} \qquad (6)$$

Fig. 3 shows the absolute equivariance measured for each augmentation. Note that this is performed on a model already trained with the usual augmentation policy containing all the augmentations. If an augmentation induces a large displacement, it means the embedding is highly sensitive to the given augmentation. What we can see from Fig. 4, is that regardless of the dataset used, the model achieves poor sensitivity to horizontal flip and grayscale. However, on ImageNet, we observe a high sensitivity to color jitter as well as medium sensitivity to crop and gaussian blur. On CIFAR10 we observe a strong sensitivity to crop and a medium sensitivity to color jitter. Therefore, we can conclude that our model truly learns an equivariance structure, and that the learned equivariance is more sensitive to some augmentation such as crop or color jitter.

### 3.2.3 INFLUENCE OF THE ARCHITECTURES

We study how architectural variations can influence our model. More precisely, we explore the impact of the architecture of the $g'_{\phi'}$, $u_\psi$ as well as the learned projection of $t$ mentioned in Sec. A.1. To this end, we train models for each architectural variation on CIFAR10, and report the top-1 accuracies under linear evaluation, the results are reported in Table 2. What we observe in Table 2a, is that the projection head of the equivariant latent space benefits from having more layers, yet this effect seems to plateau at some point. These results are in line with existing works (Chen et al., 2020a). While testing various architectures for the equivariant predictor Table 2b, we note only small performance variations, indicating that $u_\psi$ is robust to architectural changes. Finally, looking at

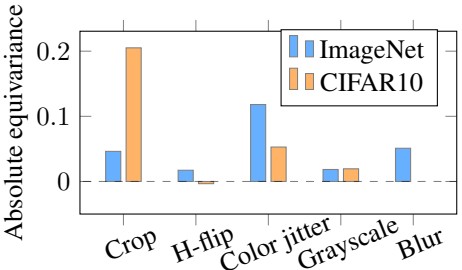 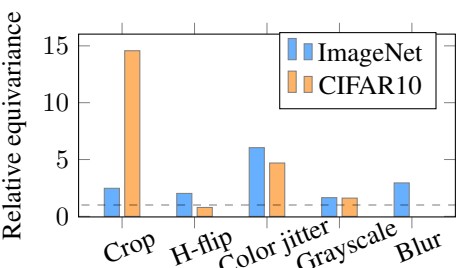

Figure 3: Absolute equivariance measure for each augmentation (the dashed line represents invariance).

Figure 4: Relative equivariance measure for each augmentation (the dashed line represents invariance).

Table 2c, we observe that removing the projection of $t$ only leads to a small drop in performance. On the contrary, complex architectures (two last lines) lead to a bigger drop in accuracy. Furthermore, while testing different output dimensions (lines 2 to 4), we note that using the same dimension for the output and for the equivariant latent space led to the highest results. Some more analysis on hyperparameter variations of our model, such as $\lambda$, batch size, or $\tau'$ can be found in Sec. A.2.

| Layers in $g'_{\phi'}$ | Top-1 |
|---|---|
| None | 88.46 |
| 1 | 91.58 |
| 2 | 92.58 |
| 3 † | 92.79 |

(a) **Equivariance projection head**

| Layers in $u_\psi$ | Top-1 |
|---|---|
| 1 † | 92.79 |
| 2 (H: 16-d) | 92.67 |
| 2 (H: 128-d) | 92.59 |
| 2 (H: 2048-d) | 92.70 |

(b) **Equivariant predictor**

| Layers in the projection of $t$ | Top-1 |
|---|---|
| None | 92.50 |
| 1 (O: 16-d) | 92.57 |
| 1 (O: 128-d) † | 92.79 |
| 1 (O: 2048-d) | 92.50 |
| 2 (H: 16-d; O: 128-d) | 92.47 |
| 2 (H: 128-d; O: 128-d) | 92.13 |
| 2 (H: 2048-d; O: 128-d) | 92.05 |

(c) **Augmentation projector**

Table 2: Top-1 accuracies (in %) under linear evaluation on CIFAR10 for some architectural variations of our module. H stands for hidden layer, O for output layer, † denotes default setup.

## 4 RELATED WORK

Most of the recent successful methods of SSL of visual representation learn a latent space where embeddings of augmentations from the same image are learned to be similar. Yet, such instance discrimination tasks admit simple constant solutions. To avoid such collapse, recent methods implement diverse tricks to maintain a high entropy for the embeddings. Grill et al. (2020) rely on a momentum encoder as well as an architectural asymmetry, Chen & He (2021) depend on a stop gradient operation, Zbontar et al. (2021) rely on a redundancy reduction loss term, Bardes et al. (2021) rely on a variance term as well as a covariance term in the loss, Chen et al. (2020a) use negative pairs repulsing sample in a batch. In this work, our task also admits collapse solutions, thus we make use of the same negative pairs as in Chen et al. (2020a) to avoid such collapse. The most recent methods are creating pairs of augmentations to maximize the similarity between those pairs. However, our addition does not rely on pairs of augmentations, and only needs a source image and an augmentation. This is similar to Misra & Maaten (2020) which requires to have a source image and an augmentation, however, they use these pairs to learn an invariance task while we use them to learn an equivariance task.

Our approach is part of a line of recent works, which try to perform additional tasks of sensitivity to augmentation while learning an invariance task. This is the case of E-SSL (Dangovski et al., 2021), which simultaneously learns to predict rotations applied to the input image while learning an invariance pretext task. This way, their model learns to be sensitive to the rotation transformation, usually

not used for invariance. Where this can be considered as a form of equivariance (a rotation in input space produces a predictable displacement in the prediction space) this is far from the equivariance we explore in this paper. Indeed, E-SSL sensitivity task can be seen as learning an instance-invariant pretext task, where for any given input, the output represents only the augmentation (rotation) used. Here, we explore equivariance sensitivity both to images and to augmentations. Moreover, we only consider sensitivity to the augmentations used for invariance. In LooC (Xiao et al., 2020), authors propose to use as many different projection heads as there are augmentations and learn each of these projection heads to be invariant to all but one augmentation. This way the projection heads can implicitly learn to be sensitive to an augmentation. Still, they do not control how this sensitivity occurs, where we explicitly define an equivariance structure for the augmentations-related information. Note that a work has tried to tackle the trade-off from the other side, by trying to reduce the shortcut learning occurring, instead of adding sensitivity to augmentations. Robinson et al. (2021) shows that shortcut learning occurring in invariant SSL is partly due to the formulation of the loss function and proposes a method to reduce shortcut learning in contrastive learning.

Some other works have also successfully used equivariance with representation learning. For instance, Jayaraman & Grauman (2015) uses the same definition of equivariance as us and successfully learns an equivariant latent space tied to ego-motion. Still, their objective is to learn embodied representations as well as using the learned equivariant space, in comparison we only use equivariance as a pretext task to learn representations. Moreover, we do not learn equivariance on the representations, but rather on a non-linear projection of the representations. Lenc & Vedaldi (2015) learns an equivariant predictor on top of representations to measure their equivariance, however, to learn that equivariance, they require the use of strong regularizations.

## 5 CONCLUSION AND PERSPECTIVES

Recent successful methods for self-supervised visual representation rely on learning a pretext task of invariance to augmentations. This encourages the learned embeddings to discard information related to transformations. However, this does not fully consider the underlying dilemma that occurs in the choice of the augmentations: strong modifications of images are required to remove some possible shortcut solutions, while information manipulated by the augmentation could be useful to some downstream tasks. In this paper, we have introduced EquiMod, a generic equivariance module that can complement existing invariance approaches. The goal of our module is to let the network learn an appropriate form of sensitivity to augmentations. It is done through equivariance via a module that predicts the displacement in the embedding space caused by the augmentations. Our method is part of a research trend that performs sensitivity to augmentations. Nonetheless, compared to other existing works, we perform sensitivity to augmentations also used for invariance, therefore reducing the trade-off, while defining a structure in our latent space via equivariance.

Testing EquiMod across multiple invariance baseline models and datasets almost always showed improvement under linear evaluation. It indicates that our model can capture more pertinent information than with just an invariance task. In addition, we observed a strong robustness of our model under architectural variations, which is a non-negligible advantage as training such methods is computationally expensive, and so does the hyperparameters exploration. When exploring the sensitivity to the various augmentations, we noticed that the latent space effectively learns to be equivariant to almost all augmentations, showing that it captures most of the augmentations-related information.

For future work, we plan on testing our module on more baseline models or even as a standalone. As EquiMod almost always improved the results in our tests, it suggests that EquiMod could improve performances on many more baselines and datasets. Then, since E-SSL adds sensitivity to rotation, yet still does not consider sensitivity to augmentations used for invariance, it would be interesting to study if combining EquiMod and E-SSL can improve even further the performances. Another research axis is to perform an in-depth study of the generalization and robustness capacity of our model. To this end, we want to explore its capacity for transferability (fine-tuning) and few-shot learning, both on usual object recognition datasets, but also on more challenging datasets containing flowers and birds as in Xiao et al. (2020). Since the trade-off theoretically limits the generalization on the learned representation, and since we reduce the effect of the trade-off, we hope that EquiMod may show advanced generalization and robustness properties. On a distant horizon, the equivariant structure learned by our latent space may open some interesting perspectives related to world model.

ACKNOWLEDGMENTS

This work was performed using HPC resources from GENCI-IDRIS (Grant 2021-AD011013160 and 2022-A0131013831) and GPUs donated by the NVIDIA Corporation. We gratefully acknowledge this support.

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

## A  APPENDIX

### A.1  ENCODING OF THE AUGMENTATIONS

We use the classical augmentations of the literature, which depend on the dataset and model used, applied in the given order:

- Resized Crop: crop a subregion of the image;
- Horizontal flip: flip the image with a given probability;
- Color jitter: jitter the image on different aspects with a random order (brightness, saturation, contrast, and hue) and with a given probability;
- Gray-scale: gray-scale the image with a given probability;
- Gaussian blur (not used with CIFAR10 except in BYOL): blur the image using a sampled $\sigma$ and with a given probability;
- Solarize (applied only with BYOL): solarize the image with a given probability.

We refer the reader to the original papers (Chen et al., 2020a; Grill et al., 2020; Zbontar et al., 2021) to know how the different methods parameterize these augmentations (e.g. values of the probability, or intervals of values sampled, as factors in color jitter).

To encode these augmentations, we represent them by a numerical vector where some of the components are binary values related to the use of augmentations (for those applied with some probability) and some others are numerical values corresponding to some parameters (of the parameterized transformations). We only consider the corresponding augmentations w.r.t. the tested dataset and model. For each of these considered augmentations except crop, we define an element valued at 1 when the augmentation is performed and valued at 0 otherwise (since each augmentation, but the crop, is applied with a given probability it may be applied or not). Then, some augmentations require additional elements. To this end, we define elements to represent these parameters using the following direct ways (note that when a parametrized augmentation is not applied due to its probability of application, its numerical components are set to some predefined default values):

- *Resized Crop* (4 elements): $x$ and $y$ coordinates of the top-left pixel of the crop as well as width and height of the crop.

- *Color Jitter* (8 elements): the jitter factors for brightness, saturation, contrast, and hue $(1, 1, 1, 0$ is the default encoding if color jitter is not applied), as well as their order of application. More precisely, to encode the order of modification, we use the following mapping $\{0 : \text{brightness}, 1 : \text{contrast}, 2 : \text{saturation}, 3 : \text{hue}\}$. For instance an encoding with "1, 3, 2, 0" would mean that contrast jitter is first applied, then hue, contrast, and finally brightness $(0, 1, 2, 3$ is the default encoding if color jitter is not applied).
- *Gaussian Blur* (1 element): the value of sigma used (0 if blur is not applied).

At this point, we have a numerical vector that represents which augmentations are applied or not and what are their parameters if any, see the following Sec. A.1.1 and Sec. A.1.2 for some examples. We then normalize this vector component-wise using experimental mean and standard deviation computed over many examples, and we use a perceptron to project the constructed vector into a 128d latent space. This perceptron is learned jointly with the rest of the model.

### A.1.1 EXAMPLE 1

Here is an example of one transformation applied during the learning of BYOL on ImageNet.

Let's consider the randomly generated transformation composed of the following augmentations:

- Crop at coordinates x, y=(12, 9) with width, height of (120, 96);
- Probabilistic horizontal flip not triggered;
- Probabilistic color jitter triggered with factors and order of: hue -0.09, contrast 1, saturation 0.84, brightness 1.13;
- Probabilistic gray-scale triggered;
- Probabilistic blur not triggered;
- Probabilistic solarize not triggered;

According to A.1, for the binary part representing the performed augmentations we have [0, 1, 1, 0, 0] for [No H-Flip, Yes Color jitter, Yes Gray-scale, No Blur, No Solarize] (one per augmentation, except crop which is always performed), and for the parameterized transformations : [12, 9, 120, 96, 1.13, 1, 0.84, -0.09, 3, 1, 2, 0, 0] for [Crop X, Crop Y, Crop Width, Crop Height, Brightness Factor, Contrast Factor, Saturation Factor, Hue Factor, Index of the First Color Modification Applied, Index of the Second Color Modification Applied, Index of the Third Color Modification Applied, Index of the Fourth Color Modification Applied, Default value for sigma (as blur is not triggered)]

Finally, this gives us the 18d vector [0, 1, 1, 0, 0, 12, 9, 120, 96, 1.13, 1, 0.84, -0.09, 3, 1, 2, 0, 0], which is then normalized and given to a perceptron to project it to a 128d vector.

### A.1.2 EXAMPLE 2

And here is another example this time with SimCLR on CIFAR10 (which uses a different augmentation policy, thus solarization and blur are not considered).

Let's consider the randomly generated transformation composed of the following augmentations:

- Crop at coordinates x,y=(1, 2) with width,height of (24, 27);
- Probabilistic horizontal flip triggered;
- Probabilistic color jitter not triggered;
- Probabilistic gray-scale not triggered;

For the binary part representing the performed augmentations, we have [1, 0, 0] for [Yes H-Flip, No Color jitter, No Gray-scale]. And for the parametrized transformations : [1, 2, 24, 27, 1, 1, 1, 0, 0, 1, 2, 3] for [Crop X, Crop Y, Crop Width, Crop Height, Brightness Factor (Default), Contrast Factor (Default), Saturation Factor (Default), Hue Factor (Default), Index of the First Color Modification (Default), Index of the Second Color Modification (Default), Index of the Third Color Modification (Default), Index of the Fourth Color Modification (Default)]. Note the default values for all the parameters of the color jitter which is not triggered.

This gives us the 15d vector [1, 0, 0, 1, 2, 24, 27, 1, 1, 1, 0, 0, 1, 2, 3], which is then normalized and given to a perceptron to project it to a 128d vector.

## A.2 Influence of hyperparameters ($\lambda$, $\tau'$ and batchsize)

In this section, similarly to Sec.3.2.3, we study how variations of minor hyperparameters can influence our model. To that purpose, we train models on CIFAR10 for each hyperparameter modification and present the top-1 accuracy under linear evaluation.

We first inspect the influence of the $\lambda$, the weighting factor between our equivariance loss and the invariance baseline loss. One can see Table 3a that when $\lambda$ is small ($< 1$) there is a drop in performance. As $\lambda$ can be seen as weighting the importance between the equivariance and the invariance terms of the loss, this confirms that our model learns better features when our equivariance addition is considered with at least the same importance as the invariance task. On the opposite, interestingly, where $\lambda$ is set to high values such as 5 or 10, we do not observe a clear modification of the performance. This tends to indicate that there is no degradation of the representation when the equivariance is prioritized.

Then we study the temperature hyperparameter of the NT-Xent loss that we use to learn equivariance. Similarly to what is reported in Chen et al. (2020a), we find Table 3b that the optimal values to be around $0.2$ and $0.5$.

Finally, we explore the impact of the batch size on the learned representations. This hyperparameter directly determines the number of negative pairs, therefore it highly influences the learning dynamic. We observe Table 3c a decrease in performance where the batch size is too small ($\leq 256$) or too big ($\geq 1024$). Once again, these findings are in line with the literature (Chen et al., 2020a).

| $\lambda$ Factor | Top-1 |
|---|---|
| 0 | 90.96 |
| 0.1 | 92.07 |
| 0.2 | 92.31 |
| 0.5 | 92.37 |
| 1 † | 92.79 |
| 2 | 92.33 |
| 5 | 92.81 |
| 10 | 92.66 |

(a) **Weighting factor between equivariance and invariance losses**

| Temperature $\tau'$ | Top-1 |
|---|---|
| 0.05 | 92.13 |
| 0.1 | 92.13 |
| 0.2 † | 92.79 |
| 0.5 | 92.31 |
| 1 | 92.14 |

(b) **Temperature of the NT-Xent used in our equivariance loss**

| Batch size | Top-1 |
|---|---|
| 64 | 92.23 |
| 128 | 92.24 |
| 256 | 92.38 |
| 512 † | 92.79 |
| 1024 | 92.23 |

(c) **Batch size**

Table 3: Top-1 accuracies (in %) under linear evaluation on CIFAR10 for some hyperparameter variations of our module. † denotes default setup.

## A.3 Additional results

The Table 4 shows the impact of the number of training epochs on the results of the linear evaluation of BYOL with and without EquiMod.

| Method | ImageNet | | CIFAR10 | |
|---|---|---|---|---|
| | Top-1 | Top-5 | Top-1 | Top-5 |
| BYOL∗ (100 epochs) | 62.09 | 84.01 | - | - |
| BYOL∗ + EquiMod (100 epochs) | **65.55** | **86.74** | - | - |
| BYOL∗ (300 epochs) | 71.34 | 90.35 | - | - |
| BYOL∗ + EquiMod (300 epochs) | **72.03** | **90.77** | - | - |
| BYOL∗ (1000 epochs) | **74.03** | **91.51** | 90.44 | 99.62 |
| BYOL∗ + EquiMod (1000 epochs) | 73.22 | 91.26 | **91.57** | **99.71** |

Table 4: **Linear Evaluation**; top-1 and top-5 accuracies (in %) under linear evaluation on ImageNet and CIFAR10 (symbols ∗ denote our re-implementations).

