# OpenReview forum: "EquiMod: An Equivariance Module to Improve Visual Instance Discrimination"
_ICLR.cc/2023/Conference — ICLR 2023 poster_

### Official Review · Reviewer_URYd · 2022-10-21

**Confidence:** 3
**Correctness:** 3
**Technical Novelty And Significance:** 4
**Empirical Novelty And Significance:** 3
**Recommendation:** 8

**Clarity, Quality, Novelty And Reproducibility:**

Quality

I am very impressed by the ideas and the writing of this paper.  The method is simple and well-motivated, and successfully makes use of the concept of equivariance within the context of SSL.  The evaluations address many aspects of the method (influence of architecture, which transformations are actually learned) that a potential use would find relevant.

Clarity

The motivation and details of the method are clearly presented.  The conceptual figures 1 and 2 also aid in grasping the method quickly.


Originality

The ideas of the paper are strikingly original.  As noted, some other works have made use of equivariance before, but the elegance of this approach surpasses them.

**Strength And Weaknesses:**

Strengths
+ Analyzes the key ideas behind data augmentation and identifies a key shortcoming of data augmentation.
+ Proposes an innovative, elegant, and general solution to overcome the identified problem of data augmentation.
+ Concept is carefully explained with aid of diagrams and appropriate mathematical notation.
+ Evaluation shows meaningful improvements achieved using method.
+ Evaluation includes an experiment that shows that Equimod indeed learns equivariances, not just invariances.
+ Evaluation includes ablation study of architectures.

Weaknesses
- Evaluations could be made stronger.  Training for only 90 or 100 iterations is too short; methods should be training until convergence for a fair comparison with and without Equimod. [addressed by rebuttal]
- Intuition for Equimod loss (eq 4) not fully explained.  How does the loss differ in the case of a positive pair vs a negative pair?  And how are indices i and j sampled (e..g uniformly at random)? [addressed by rebuttal]
- Not clear how to tune the scaling of the Equimod loss.  The paper could be improved if the choice of lambda could be explored.  How is lambda set for the experiments?  Could an ablation on lambda (setting to different values and seeing the performance) be conducted? [addressed by rebuttal]
- Relatedly, the theory for how well Equimod might work to avoid "shortcut solutions" and optimally trade off is totally heuristic at the moment.  We see that Equimod works, but the paper doesn't give us a clear idea of the details of how that is possible theoretically.  Perhaps a theoretical analysis of a toy model would help shed some light on the success of the method.
- What is the computational cost of Equimod? [addressed by rebuttal]
- Will the code be shared? [addressed by rebuttal]

**Summary Of The Paper:**

Proposes a modification of the data-augmentation approach to self-supervised learning where the parameters of the transformation used in data-augmentation can be used by the network to predict the latent variables of the un-augmented input.  This is advantageous when there exists a set of input features (such as color histogram) which would potentially cause the network to converge to bad local minima (shortcut solutions), yet could still be useful for classification.  Previous approaches would use data augmentations to enforce invariance to those features, which avoids the bad local minima but at the cost of losing the ability to use those features for classification.  Equimod therefore improves performance by combining a loss that encourages invariance to data augmentations and a loss that encourages the learning of a representation that is equivariant to data augmentations.  This is hypothesized to allow the network to avoid bad local minima without becoming totally invariant to useful features that are perturbed by data augmentations.  Experimental results show improved performance of SSL methods by modifying them to include the Equimod sub-network for predicting the effect of data augmentations on the latent space.

**Summary Of The Review:**

The paper presents a very interesting method for the field of SSL, and after rebuttal, provides more clarity and insight into the proposed method.  While a better evaluation (e.g. on problems such as flowers or trees) might allow the work to have a greater impact, I  think in its present form it will be useful for SSL researchers.

---

> ### Author Response · Authors · 2022-11-10
> **Answer to Reviewer URYd (1/2)**
>
> Dear reviewer,
>
> First of all, we would like to thank you for the time you took to review our article. We would also like to thank you for all the constructive remarks you made that will help us to improve our work. Finally, thank you for your enthusiasm for our work, we appreciate it. We pointed out in some of our responses that modifications will be made on the next version of the article. We are currently working on it and will make it available to you at last November 14 with the list of modifications made.
>
> ---
> > Evaluations could be made stronger. Training for only 90 or 100 iterations is too short; methods should be training until convergence for a fair comparison with and without Equimod.
>
> 90 epochs are used only for training the linear classifier of the final evaluation. With these 90 epochs, the classifier fully converged. This is the usual training procedure used in the literature, for instance in SimCLR (Chen et al. (2020a)).
>
> SimCLR and Barlow Twins were trained with the original number of epochs mentioned in the articles (800 epochs). Regarding the training of BYOL during only 100 epochs, this was unfortunately due to our limited access to computational resources. Since we were not able to do a full 1000 epochs run as originally made by Grill et al. (2020), we decided to do as the authors of the E-SSL paper (Dangovski et al., 2021), who also had limited resources, to have a fair comparison on this limited setup of 100 epochs. In the meantime, we obtained the results for the learning during 300 epochs (71.34% without EquiMod and 72.03% with EquiMod) which show the same tendency. The training with 1000 epochs are currently running and we hope to obtain them before November 18. Otherwise, these results will be made available in the camera-ready version (if requested) and on the GitHub release.
>
> ---
> > Intuition for Equimod loss (eq 4) not fully explained. How does the loss differ in the case of a positive pair vs a negative pair? And how are indices i and j sampled (e..g uniformly at random)?
>
> As stated in the article, our module learns to predict the displacement in the embedding space caused by the augmentations. This prediction is performed by our predictor, which outputs $\hat{z}'_i$ for one given image and one augmentation. On the other hand, we can compute the embedding of the augmented image, which we note $z'_i$. Since we consider the case of equivariance, we want the application of the transformation via a predictor in the latent space ($\hat{z}'_i$) to be made similar to applying the transformation in the input space and then computing the embedding ($z'_i$).
>
> Still, this formulation alone allows collapse solutions, thus we use a classical trick of contrastive methods (as SimCLR) by using negative pairs to repulse embeddings from different images (here in the sense of instance, not views) present in the batch.
>
> Since we place ourselves as a complement to existing invariant methods, which learn similar embeddings for two augmentations of the same image (a.k.a. views, noted $v_i$ and $v_j$), our batch contains pairs of embeddings coming from the same image. For this reason, our loss needs $j$ to avoid the repulsion of $z'_i$ and $z'_j$ as these embeddings are originating from the same image.
>
> Note that SimCLR does not explicitly remove this repulsion of (i, j) because it is canceled by the attraction of (i, j) in their numerator, yet in our case, we do not attract the pair (i, j) thus we simply remove this repulsion in the denominator.
>
> ---
> > Not clear how to tune the scaling of the Equimod loss. The paper could be improved if the choice of lambda could be explored. How is lambda set for the experiments? Could an ablation on lambda (setting to different values and seeing the performance) be conducted?
>
> Thank you for pointing this out. We forgot to mention that lambda is set to 1 for all our experiments. We already have the ablation study for lambda on our side, it just reveals that a small value (<1) for lambda leads to slightly sub-optimal performances, we will include this in the appendix.
>
> Here are the results of our ablation on the $\lambda$ parameter (using a similar setup as the ones already used for other ablation experiments):
> - lambda = 0.1 -> 92.07% top-1 acc on CIFAR10
> - lambda = 0.2 -> 92.31% top-1 acc on CIFAR10
> - lambda = 0.5 -> 92.37% top-1 acc on CIFAR10
> - lambda = 1.0 -> 92.79% top-1 acc on CIFAR10
> - lambda = 2.0 -> 92.33% top-1 acc on CIFAR10
> - lambda = 5.0 -> 92.81% top-1 acc on CIFAR10
> - lambda = 10. -> 92.66% top-1 acc on CIFAR10
>
> **(1/2) More in other comments due to the character limit.**

---

> > ### Author Response · Authors · 2022-11-10
> > **Answer to Reviewer URYd (2/2)**
> >
> > > Relatedly, the theory for how well Equimod might work to avoid "shortcut solutions" and optimally trade off is totally heuristic at the moment. We see that Equimod works, but the paper doesn't give us a clear idea of the details of how that is possible theoretically. Perhaps a theoretical analysis of a toy model would help shed some light on the success of the method.
> >
> > Indeed, this would be a great idea, we did not think of. This will be considered for future work, thank you.
> >
> > ---
> > > What is the computational cost of Equimod?
> >
> > From an analytical point of view, it requires computing the representations (via a ResNet) for a 3rd image, rather than for only 2 views, which is a +50% overhead. Over this, we add the overhead caused by the equivariance projection head, as well as the computation for our equivariance loss. Yet, relative to the ResNet computation, the projection head and loss computations are negligible. By comparison, other equivariance methods perform much worse, for instance, E-SSL requires 4 additional images for the four-fold rotation (+200% overhead), and LooC needs N+1 views, where N is the number of augmentations, (thus +250% using 6 augmentations as in our worst case). Furthermore, for invariance methods, one of the bottlenecks is to apply the augmentations as it cannot be done on GPUs. The advantage of EquiMod is that the additional image required is non-augmented.
> >
> > Unfortunately, we cannot provide an execution time difference between our method and invariance ones, as for our experimentations with invariance methods, we have simply set $\lambda$ to 0 (i.e. ponderating the equivariance loss to 0, thus keeping the overhead). This choice guarantees that there is no implementation difference between our experimentations with and without EquiMod. Nonetheless in practice, using highly parallelized hardware, we observe that EquiMod produces almost no time overhead, as we perform 800 epoch training in 6 days of computation, which is similar to invariance methods claiming 6 days/1 week (using fairly similar hardware).
> >
> > More importantly, the overhead that may exist is only during the training phase, but at inference, we have the same amount of computation as other approaches.
> >
> > ---
> > > Will the code be shared?
> >
> > As said in the footnote Page 3, code will be made available on GitHub before the camera-ready deadline, as well as model checkpoints to allow for reproducibility.
> >
> > ---
> >
> > We thank you for your time and understanding,
> >
> > Best.

---

> > ### Comment · Reviewer_URYd · 2022-11-16
> > **more details about batch?**
> >
> > > Since we place ourselves as a complement to existing invariant methods, which learn similar embeddings for two augmentations of the same image (a.k.a. views, noted  and ), our batch contains pairs of embeddings coming from the same image. For this reason, our loss needs  to avoid the repulsion of
> >  and
> >  as these embeddings are originating from the same image.
> >
> > Thanks, this makes sense.  I'm assuming that a batch consists of $2N$ images, $N$ original images and $N$ augmentations?  It might help to mention this in the paper.  Also, if we number the original images $i=1,...,N$, we could number the augmentations as $j=N+1,...,2N$ and we have pairs $(i, j) = (i, N+i)$.  Is that right?  Again, it might help the reader to make this explicit.

---

> > > ### Author Response · Authors · 2022-11-18
> > > **Re: more details about batch?**
> > >
> > > Dear reviewer,
> > >
> > > We thank you for taking the time to respond to our comments. We also appreciate your consideration of upgrading your rating.
> > >
> > > Considering $N$ original images, for each of the image $x$, there are two different augmentations that produce the views $v_i$ and $v_j$, thus resulting in $2N$ augmented images. The most recent invariance methods, the ones we complement in the paper, learn from these $2N$ augmented images by maximizing the similarity between embeddings of views coming from the same source image ($z_i$ and $z_j$). So the positive pairs are $i$ and $i+N$ (resp. $i+N$ and $i$), but $i$ does not correspond to the original image but to an augmentation of the image. By the way, this makes $2N$ invariance positive pairs as the order count ($(i, j) \neq (j, i)$). Note that it follows the invariance described in Eq. 2.
> > >
> > > What you mention in your comment corresponds to Eq. 1, where invariance is performed between the source image and its view (rather than two views). Note that it is the invariance learned by PIRL (Misra & Maaten (2020)).
> > >
> > > Following your remark, we updated our paper by being more specific about how the batch is built (See the beginning of the second paragraph in Sec. 2.2 "Method").
> > >
> > > Once again, we thank you for your time and understanding,
> > >
> > > Best.

---

> > ### Comment · Reviewer_URYd · 2022-11-16
> > **ablation of lambda helps**
> >
> > > We already have the ablation study for lambda on our side, it just reveals that a small value (<1) for lambda leads to slightly sub-optimal performances, we will include this in the appendix.
> >
> > Thanks for looking into this!  It raises my evaluation of the paper.  (score to be updated after discussion period).

---

### Official Review · Reviewer_9hFv · 2022-10-25

**Confidence:** 4
**Correctness:** 2
**Technical Novelty And Significance:** 2
**Empirical Novelty And Significance:** 2
**Recommendation:** 3

**Clarity, Quality, Novelty And Reproducibility:**

The clarity and quality of the writing are generally okay, but need improvement. The idea is relatively new to self-supervised learning, but some key components were not well clarified with sufficient details and justification. The general idea is not hard to reproduce, but it may not be easy to reproduce the exact same architecture/results due to the lack of some technical details (e.g. the encoding scheme and the detailed network architectures).

**Strength And Weaknesses:**

**Strengths**

\+ The idea of equivariance in the feature space to help self-supervised representation learning is interesting.

\+ The proposed method was shown to be effective on linear evaluation (Table 1), by showing a performance gain when added to some prior works.


**Weaknesses**

\- The statement "These methods rely on maximizing the similarity between..." in the Abstract is inaccurate. Not all self-supervised visual representation learning methods are based on such a contrastive learning scheme. Instead, most of the early methods are based on pretext tasks. Suggest revising the corresponding claims and also the title.

\- The main idea of this work is based on the assumption of the self-supervised visual learning method being contrastive-based ones. Whereas there are quite a few non-contrastive learning approaches. As a result, the main contributions of this work may be a bit limited.

\- It is unclear how the defined u_\psi was guaranteed to represent the transformation t. From the description, t was encoded by another set of layers, which are also learnable, making the equivariance here (esp. the "equivariant predictor") a bit misleading.
The encoding of the transformations (Sec. 2.3.2) is rather confusing. In the beginning, it looks like the authors used one-hot encoding, but later some of the augmentations directly use the coordinates or real values (i.e. 1, 2, 3). The generalization for the augmentation set, as a result, is another issue. What if new augmentations were included? How would they be encoded?

\- It is unclear why the final model still needs the conventional invariance loss (i.e. the first term in the final loss function), if the proposed new equivariance-based loss is as claimed to be effective. Jointly optimizing both invariance and equivariance also seems to be confusing. How does the model actually learn in this case?

\- It is unclear why the Barlow Twins was only evaluated on the small-scale dataset CIFAR10.

\- The quality of the learned representation was only evaluated on the linear evaluation setting, which is a bit insufficient to get a clear conclusion. There are quite a few other downstream tasks that could be used as reported in the literature (e.g. fine-tuning, detection, and segmentation to name a few).

\- The proposed method was motivated by the case where "augmentation information is useful" as claimed by the authors, and example cases are flowers and birds (as claimed in the Introduction). But this was not validated in the experiment. There are some fine-grained datasets for such categories (e.g. the Oxford 102 flowers dataset and the Caltech-UCSD Birds-200 dataset) that should have been used to validate the claims.

\- It is unclear why not use an objective similar to Eq. (5) or (6) to constrain the equivariance model training, but instead only maximize the similarity between z_i' and \hat{z}_i'?

\- Would an absolute value makes more sense for Eq (5)? Otherwise, what does a negative value mean, as the second case "H-flip" shown in Fig. 3?

\- Line below Eq. (5), "Fig. 4" should be "Fig. 3".

**Summary Of The Paper:**

This paper presented a new module to improve self-supervised contrastive visual representation learning. Specifically, the proposed module focused on equivariance in the leaned latent space. Experimental analysis on two public datasets showed that when applying the proposed module to existing methods (SimCLR, BYOL and Barlow Twins) the performance was improved. The main contribution of this paper is the proposed equivariance module.

**Summary Of The Review:**

The main idea of this work about the equivariance in latent space is interesting and may have the potential to explore and contribute to the community. The shown experiments also suggested the effectiveness of the proposed method to some extent. But some technical design was not clearly presented without sufficient evidence to back them up. There are also several unclear claims and statements. The experiments are also insufficient to validate the claims. As a result, I would suggest the authors revise their paper accordingly for a future submission.

---

> ### Author Response · Authors · 2022-11-10
> **Answer to Reviewer 9hFv (1/?)**
>
> Dear reviewer,
>
> First of all, we would like to thank you for the time you took to review our article. We also want to thank you for all the constructive remarks you made that will help us to improve our work. We pointed out in some of our responses that modifications will be made on the next version of the article. We are currently working on it and will make it available to you at last November 14 with the list of modifications made.
>
> ---
> > The statement "These methods rely on maximizing the similarity between..." in the Abstract is inaccurate. Not all self-supervised visual representation learning methods are based on such a contrastive learning scheme. Instead, most of the early methods are based on pretext tasks. Suggest revising the corresponding claims and also the title.
>
> The aim of this article is to improve recent self-supervised visual representation learning methods that rely on invariance pretext tasks, which include contrastive and non contrastive approaches. Self-supervised learning is based on the use of pretext tasks, which can be contrastive or not as you mentioned it. Currently, the most efficient methods for self-supervised visual representation learning are mostly contrastive ones. Most state-of-the-art approaches for visual instance discrimination are contrastive (i.e. have positive and negative pairs), though not all (like BYOL (Grill et al. (2020)), for example) are. In the literature, some papers (for instance in VICReg (Bardes et al. (2021)) use self-supervised learning, contrastive learning and visual instance discriminative almost as synonymous (what we did here), which indeed can be misleading as your review pointed out. Therefore, in the next version of our article, we will be more precise about the terms used. It will include an update version of the title that will be “EquiMod: An Equivariance Module to Improve Visual Instance Discrimination”
>
> ---
> > The main idea of this work is based on the assumption of the self-supervised visual learning method being contrastive-based ones. Whereas there are quite a few non-contrastive learning approaches. As a result, the main contributions of this work may be a bit limited.
>
> The scope of this article is to push recent state-of-the-art methods, which as mentioned above are mostly represented by instance discrimination since SimCLR (Chen et al. (2020a)), and have witnessed increasing popularity since then (Caron et al., 2020; Chen et al., 2020a;b; Chen & He, 2021; Bardes et al., 2021; Grill et al., 2020; He et al., 2020; Misra & Maaten, 2020; Zbontar et al., 2021). Therefore, we do not consider improving methods such as four-fold rotations, image enhancing, colorization, jigsaw, or reconstruction, which are out of the scope of our article.
>
> We are not stating to improve all existing self-supervised learning methods, we are claiming to push actual state-of-the-art by using our generic module. By precising the terminology used (as stated in the previous point), this will be made clearer in the next version of the article.
>
> ---
> > It is unclear how the defined u_\psi was guaranteed to represent the transformation t. From the description, t was encoded by another set of layers, which are also learnable, making the equivariance here (esp. the "equivariant predictor") a bit misleading. The encoding of the transformations (Sec. 2.3.2) is rather confusing. In the beginning, it looks like the authors used one-hot encoding, but later some of the augmentations directly use the coordinates or real values (i.e. 1, 2, 3). The generalization for the augmentation set, as a result, is another issue. What if new augmentations were included? How would they be encoded?
>
> Our predictor $u_\psi$ learns to predict the displacement in the embedding space caused by the augmentations. In other words, for one given image embedding and one transformation, it predicts the embedding of the transformed image. This relates to equivariance as it is the same principle as $u_t$ in Eq 3. Applying the transformation via a predictor in the latent space ($\hat{z}'_i$) should be made similar to applying the transformation in the input space and then computing the embedding ($z'_i$). Therefore, we maximize the similarity between $\hat{z}'_i$ and $z'_i$. However, to be able to input the transformation (i.e. a function) in our neural network, we represent $t$ via a numerical representation, potentially learned, of the parameters that define it (cf section 2.3.2). To sum up, $u_\psi$ is a parametrized version of $u_t$ as it represents the transformation of the embedding caused by the augmentation $t$, where $t$ is encoded as a learned projection of the parameters defining completely the augmentation.
>
> **(1/?) More in other comments due to the character limit.**

---

> > ### Author Response · Authors · 2022-11-10
> > **Answer to Reviewer 9hFv (2/?)**
> >
> > For the encoding of the transformations, to first clarify, there is no one-hot encoding as we represent all augmentations applied to an image in a single numerical vector. To construct this vector, some of the components are binary values related to the use of an augmentation or numerical values corresponding to some parameters (of the parametrized transformations). For each of the augmentations, we first define an element valued at 1 when the augmentation is performed and valued at 0 otherwise (since each augmentation is applied with a given probability it may be applied or not). Then, some augmentations require additional elements, such as crop, color jitter, and blur which are parametrized respectively by the region of the crop, the jitter factors and order, and the sigma used to be fully represented. To this end, we propose direct ways of representing such parameters of the augmentations (Sec 2.3.2), this gives us respectively 4, 8, and 1 components to represent the parameters of these three augmentations. Note that when a parametrized augmentation is not applied (due to its probability of application), its numerical components are set to some default values as detailed in (Sec 2.3.2). At this point, we have a numerical vector that represents which augmentations are applied or not and what are their parameters. Finally, we normalize this vector component-wise using experimental mean and std computed over many examples, and we use a (multi-layer) perceptron to project the constructed vector into a latent space (this (multi-layer) perceptron is simply learned jointly with the rest of the model). By the way, regarding table 2.c, this last projection step can be removed without a large impact on the performance.
> >
> > Besides depending on the model and the dataset, the number of applied transformations may vary. For instance, solarization is only used in BYOL. Thus, the size of the parameters vector is adapted to each protocol by simply removing unused dimensions w.r.t. the tested setup. This point will be added in section 2.3.2 of the next version of the article.
> >
> > Moreover, your review made us realize that examples can be useful to help its understanding, so we will add some in the appendix of the next version, similar to the two following examples.
> >  Here is an example of one transformation applied during the learning of BYOL on ImageNet:
> > Considering the randomly generated transformation composed of the following augmentations:
> > - Crop at coordinates x,y=(12, 9) with width,height of (120, 96);
> > - Probabilistic horizontal flip not triggered;
> > - Probabilistic color jitter triggered with factors and order of: hue -0.09, contrast 1, saturation 0.84, brightness 1.13;
> > - Probabilistic gray-scale triggered;
> > - Probabilistic blur not triggered;
> > - Probabilistic solarize not triggered;
> >
> > For the elements valued at 1 when augmentation is performed and valued at 0 otherwise, we have [0, 1, 1, 0, 0] for [No H-Flip, Yes Color jitter, Yes Gray-scale, No Blur, No Solarize] (one per augmentation, except crop which is always performed). And for the parametrized transformations, according to (Sec 2.3.2) : [12, 9, 120, 96, 1.13, 1, 0.84, -0.09, 3, 1, 2, 0, 0] for [Crop X, Crop Y, Crop Width, Crop Height, Brightness Factor, Contrast Factor, Saturation Factor, Hue Factor, Index of the First Color Modification Applied, Index of the Second Color Modification Applied, Index of the Third Color Modification Applied, Index of the Fourth Color Modification Applied, Default value for sigma (as blur is not triggered)]
> >
> > Finally, this gives us the 18d vector [0, 1, 1, 0, 0, 12, 9, 120, 96, 1.13, 1, 0.84, -0.09, 3, 1, 2, 0, 0], which is then normalized and given to a perceptron to project it to a 128d vector (note that we also tested other settings for this last projection, see table 2.c).
> >
> > And here is another example this time with SimCLR on CIFAR10 (which uses a different augmentation policy):
> > Considering the  randomly generated transformation composed of the following augmentations:
> > - Crop at coordinates x,y=(1, 2) with width,height of (24, 27);
> > - Probabilistic horizontal flip triggered;
> > - Probabilistic color jitter not triggered;
> > - Probabilistic gray-scale not triggered;
> >
> > For the elements valued at 1 when augmentation is performed and valued at 0 otherwise, we have [1, 0, 0] for [Yes H-Flip, No Color jitter, No Gray-scale]. And for the parametrized transformations : [1, 2, 24, 27, 1, 1, 1, 0, 0, 1, 2, 3] for [Crop X, Crop Y, Crop Width, Crop Height, Default Brightness Factor, Default Contrast Factor, Default Saturation Factor, Default Hue Factor, Index of the Default First Color Modification Applied, Index of the Default Second Color Modification Applied, Index of the Default Third Color Modification Applied, Index of the Default Fourth Color Modification Applied]. Note the default values for all the parameters of the color jitter which is not triggered.
> >
> > **(2/?) More in other comments due to the character limit.**

---

> > > ### Author Response · Authors · 2022-11-10
> > > **Answer to Reviewer 9hFv (3/4)**
> > >
> > > This gives us the 15d vector [1, 0, 0, 1, 2, 24, 27, 1, 1, 1, 0, 0, 1, 2, 3], which is then normalized and given to a perceptron to project it to a 128d vector (note that we also tested other settings for this last projection, see table 2.c).
> > >
> > > As mentioned previously, following state-of-the-art methods, we use different augmentation sets on ImageNet and on CIFAR10 (we do not use blur on CIFAR10). Additionally, BYOL uses the solarization augmentation. To adapt to such variations in the set of augmentation, the size of the parameters vector is adapted to each protocol by simply removing unused dimensions. Thus, our model was able to adapt to the (main) existing variations of the transformation currently used in the literature.
> > >
> > > By the way, as mentioned in the article, the code will be released on GitHub, which will help with the technical implementation of the encoding.
> > >
> > > ---
> > > > It is unclear why the final model still needs the conventional invariance loss (i.e. the first term in the final loss function), if the proposed new equivariance-based loss is as claimed to be effective. Jointly optimizing both invariance and equivariance also seems to be confusing. How does the model actually learn in this case?
> > >
> > > We claim the equivariance-based loss to be effective as a complement to invariance loss to avoid losing information related to augmentation, such as color. We want to combine the advantages of both approaches, i.e. that the invariance task aims for capturing features insensitive to augmentations relevant to class identification, while the equivariance task aims for extracting features sensitive to augmentations to obtain richer representation.
> > >
> > > We are able to jointly optimize both tasks thanks to the dedicated projection heads. Instead of learning a representation space that is both invariant and equivariant, we learn a projection to an invariant latent space and another projection to an equivariant one. This encourages the model to learn representations containing both features invariant and equivariant to augmentations.
> > >
> > > ---
> > > > It is unclear why the Barlow Twins was only evaluated on the small-scale dataset CIFAR10.
> > >
> > > Unfortunately, we only have access to limited computational resources, this is why we had to restrict the experimentations to the most relevant ones. On one hand, we retained SimCLR as it represents a contrastive approach to instance discrimination and performs well on CIFAR. And on the other hand, BYOL offers a different kind of architecture (as it is a bootstrapping approach rather than a contrastive one), while having the highest top-1 accuracy with linear evaluation on ImageNet using a ResNet50 backbone in a self-supervised fashion. We also tested on Barlow Twins as it is not exactly a contrastive approach nor a bootstrapping one to illustrate the generality of our approach, but limited the test on CIFAR as it is computationally reasonable to train on this dataset contrary to ImageNet.
> > >
> > > ---
> > > > The quality of the learned representation was only evaluated on the linear evaluation setting, which is a bit insufficient to get a clear conclusion. There are quite a few other downstream tasks that could be used as reported in the literature (e.g. fine-tuning, detection, and segmentation to name a few).
> > >
> > > As explained in more detail to review 1, since we have limited access to computational resources, we focused on linear evaluation which is the most widespread metric in the literature to compare our work.
> > >
> > > ---
> > > > The proposed method was motivated by the case where "augmentation information is useful" as claimed by the authors, and example cases are flowers and birds (as claimed in the Introduction). But this was not validated in the experiment. There are some fine-grained datasets for such categories (e.g. the Oxford 102 flowers dataset and the Caltech-UCSD Birds-200 dataset) that should have been used to validate the claims.
> > >
> > > We agree that the way we formulated it can be confusing, and we will remove this sentence in the next version of the article. We just wanted to say that color information is useful in representations, as it has been shown (for instance by LooC (Xiao et al., 2020)) on flowers or birds datasets.
> > >
> > > **(3/4) More in other comments due to the character limit.**

---

> > > > ### Author Response · Authors · 2022-11-10
> > > > **Answer to Reviewer 9hFv (4/4)**
> > > >
> > > > > It is unclear why not use an objective similar to Eq. (5) or (6) to constrain the equivariance model training, but instead only maximize the similarity between z_i' and \hat{z}_i'?
> > > >
> > > > Maximizing Eq 5 or 6 would mean maximizing the similarity between $z'_i$ and $\hat{z}'_i$, as well as minimizing the similarity between $z'_i$ and $z'_o$. While the first term is part of what we propose, the second one would minimize the similarity between the embedding of the view and the embedding of the non-augmented image, which may not be desirable. Indeed, this would encourage a collapse solution where the model may put on one side of the hypersphere the images not augmented, and on the other side, the images augmented (predictor simply being $u_\psi(z') = -z'$), without any form of granularity in the augmented images, which could lead in poor representation of the augmentation-related information.
> > > >
> > > > As we want an equivariant space, applying the transformation via a predictor in the latent space ($\hat{z}'_i$) should be made similar to applying the transformation in the input space and then computing the embedding ($z'_i$). That is why we maximize the similarity between $\hat{z}'_i$ and $z'_i$, while we also have negative pairs which help in not collapsing. We develop more on the reason for choosing this loss function in a response to reviewer 3 (remark 2).
> > > >
> > > > The goal of these metrics is to verify that our model is truly learning equivariance and not a simple invariance, i.e. a predictor that converged to identity. What we want is a value higher than 0 for Eq 5 and higher than 1 for Eq 6, still, we do not aim for the highest possible value, which would mean that our predictor has converged to $u_\psi(z'_i) = -z'_i$ which is almost as trivial as the identity, and thus would poorly structure our latent space.
> > > >
> > > > ---
> > > > > Would an absolute value makes more sense for Eq (5)? Otherwise, what does a negative value mean, as the second case "H-flip" shown in Fig. 3?
> > > >
> > > > Eq 5 shows the displacement caused by the predictor toward the real expected embedding of the view $v_i$. A negative displacement is thus a displacement in the opposite direction, in other words, it means that the predictor performed worse than the identity function. Still, for "H-flip", the value is close to 0, therefore, it should rather be interpreted as: the model did not learn sensitivity to "H-flip", and is thus mostly invariant to this augmentation.
> > > >
> > > > ---
> > > > > Line below Eq. (5), "Fig. 4" should be "Fig. 3".
> > > >
> > > > Thank you for pointing this out, we will fix it in the next revision.
> > > >
> > > > ---
> > > >
> > > > We thank you for your time and understanding,
> > > >
> > > > Best.

---

> > ### Comment · Reviewer_9hFv · 2022-12-13
> > **After rebuttal**
> >
> > Thanks to the authors' responses, which addressed some of the concerns. However, there are still some issues not well addressed, mainly about the claims and statements lacking convincing justification. The proposed method is based on several claims and assumptions but they are not backed up with convincing justifications. In the responses, the authors tried to explain some of them, which is good to see, but still not providing either theoretically or experimentally convincing justifications to support them. For example, the joint optimization of the invariance and equivariance, the learnable transformation together with the claimed equivariance, and the proposed equivariance metrics, to name a few. The insufficient experiment is another weakness of this paper. It is understandable of the authors-mentioned access to limited computational resources, but the claimed conclusion needs to be backed up at least by sufficient experimental validations, if not theoretically.
> >
> > As a result, this could be a good work, but the paper in its current form still prevents me from recommending a clear "accept" for ICLR and the general readers. But I have to say the rebuttal did address some of the issues and I would increase my score a bit if there is one between the 'borderline' and 'reject'. I would suggest the authors revise and improve the paper to make it stronger and consider re-submit to a future venue.

---

> > > ### Author Response · Authors · 2022-12-13
> > > **Re: After rebuttal**
> > >
> > > Dear reviewer,
> > >
> > > We thank you for taking the time to respond to our comments. We also appreciate your consideration of upgrading your rating.
> > >
> > > To make it clear, we claim to push actual state-of-the-art instance discrimination by using our generic equivariance module. We have validated EquiMod over multiple datasets and baselines, and while we agree that using more evaluation methods could make the claims stronger, we decided to focus on the main evaluation procedure of the domain to validate our claims, as we think that they can be useful to the community.
> > >
> > > This is what we try to make explicit in the revised version of the paper. If there was time remaining we would have been happy to improve our paper. Unfortunately regarding the deadline, this will not be possible.
> > >
> > > Once again, we thank you for your time and understanding,
> > > Best.

---

### Official Review · Reviewer_A3J1 · 2022-10-27

**Confidence:** 4
**Correctness:** 4
**Technical Novelty And Significance:** 3
**Empirical Novelty And Significance:** 3
**Recommendation:** 8

**Clarity, Quality, Novelty And Reproducibility:**

The paper is easy to follow, clearly written, and the method is novel. I imagine that it will be easy to reproduce given the details in the paper.

**Strength And Weaknesses:**

Strengths:

1. A novel approach to enforcing equivariance is presented. The regularizer and modification to regular architectures is simple and effective.

2. The performance of the technique is shown on multiple losses and

Weaknesses/Questions:

1. In Eq. (4) why is j not shown on the numerator?

2. Is the only difference between the equivariance projection head and equivariant predictor the fact that the augmentation parameters are fed into the predictor? Otherwise equivariance projection head and predictor could have just been merged? Is equivariance projection head used elsewhere?

3. In Figure 3 and 4, what is the value of these measures for a regularly trained (invariant) model? This will tell us the benefit of adding the regularizer.

4. What is the reason that equivariance is stronger for color compared to other augmentations for ImageNet? Any insight into this behavior?

5. Did you try any finetuning experiments?

**Summary Of The Paper:**

This paper proposes an equivariance regularizer as a modification to the usual invariance-inducing self-supervised losses. This is an interesting approach to enabling equivariance as there is no need to have a special architecture as prior work. The authors are able to train multiple self-supervised losses on a standard ResNet-50 and show good linear probe improvements over invariant baselines.

**Summary Of The Review:**

Given the overall novelty and clean/simple idea which seems to work well, I recommend acceptance.

---

> ### Author Response · Authors · 2022-11-10
> **Answer to Reviewer A3J1**
>
> Dear reviewer,
>
> First of all, we would like to thank you for the time you took to review our article. We also want to thank you for all the constructive remarks you made that will help us to improve our work. We pointed out in some of our responses that modifications will be made on the next version of the article. We are currently working on it and will make it available to you at last November 14 with the list of modifications made.
>
> ---
> > In Eq. (4) why is j not shown on the numerator?
>
> As stated in the article, our module learns to predict the displacement in the embedding space caused by the augmentations. This prediction is performed by our equivariance predictor, which outputs $\hat{z}'_i$ for one given image and one augmentation. On the other hand, we can compute the embedding of the augmented image, which we note $z'_i$. Since we consider the case of equivariance, applying the transformation via a predictor in the latent space ($\hat{z}'_i$) should be made similar to applying the transformation in the input space and then computing the embedding ($z'_i$). Therefore, we maximize the similarity between $\hat{z}'_i$ and $z'_i$, which does not involve using $j$ (which stands for another view of the image using a different augmentation). However, it is worth noting that the loss is applied to all pairs in the batch. Therefore, at some point, indexes will be $l_ji$, which attracts $\hat{z}'_j$ and $z'_j$ (and in this case, we ignore $i$ in the numerator).
>
> ---
> > Is the only difference between the equivariance projection head and equivariant predictor the fact that the augmentation parameters are fed into the predictor? Otherwise equivariance projection head and predictor could have just been merged?
>
> There is more difference than that, each of these two modules has its role:
> - Our equivariance projection head ($g’_{\phi’}$ in Figure 2) is similar to the projection head in invariance methods. It is a learned projection to construct embedding ($z'$) for a given representation ($h$). Its purpose is to avoid learning the pretext task directly on the representations, which could harm their separability.
> - On the other hand, our predictor ($u_\psi$ in Figure 2) takes an embedding (produced by the equivariance projection head) and an augmentation as inputs and predicts the displacement in the embedding space induced by the augmentation.
>
> Regarding the architecture, from the ablation study, one can see that the projector is sensible to change of depth in the architecture (Table 2.a), while the predictor is much more flexible (Table 2.b).
>
> Still, exploring ways of merging the two may be an interesting perspective.
>
> ---
> > Is equivariance projection head used elsewhere?
>
> The equivariance projection head is only used to avoid learning the equivariance task directly on the representation. This shows better results (see Table 2.a). Please note that our observations about the usage of the projection head are in line with the findings on invariance methods.
>
> ---
> > In Figure 3 and 4, what is the value of these measures for a regularly trained (invariant) model? This will tell us the benefit of adding the regularizer.
>
> While this would be very informative to further quantify the addition of our model, the proposed measures cannot be applied directly and we do not find a relevant way to adapt them. Here, in Figures 3 and 4, we study the displacement caused by the predictor (either in absolute displacement or in relative displacement), yet there is no equivalent of the predictor in invariance methods. A way to adapt could be to consider the invariance as the special case of equivariance where the predictor is the identity function, then we would have $ \hat{z}_i = z_o$. However, Eq 5 will then always be 0 and Eq 6 will be 1 (which is indeed what is represented by dashed lines in the figures). To avoid this issue, we can only consider the first term of Eq 5 ($sim(z'_i, \hat{z}'_i)$), but in this case, the value will not be comparable to Figures 3 and 4, as it would be a similarity between two embeddings and not a displacement.
>
> ---
> > What is the reason that equivariance is stronger for color compared to other augmentations for ImageNet? Any insight into this behavior?
>
> Unfortunately, we have no insight into this behavior yet. This might be related to the distribution of features sensible to each augmentation in the dataset, but this is only a very preliminary hypothesis at this time. Indeed, this is definitively a perspective for future work.
>
> ---
> > Did you try any finetuning experiments?
>
> As we have limited access to computational resources, we focused on linear evaluation which is the most widespread metric in the literature to compare our work. But as mentioned in the perspectives, we want to perform later an in-depth study of the generalization and robustness capacities of our model, which will include performing finetuning experiments.
>
> ---
>
> We thank you for your time and understanding,
>
> Best.

---

### Author Response · Authors · 2022-11-15
**Changes of the v2**

Dear Reviewers,

Once again, we thank you for your remarks, this allowed us to improve our article by performing the following modifications:
- Improving the precision of the claims and positioning: Title / Abstract / Introduction paragraph 2 / Related work paragraph 1 / Conclusion and perspectives paragraph 1;
- Clarification of the objective of our module: Method paragraph 1;
- Clarification of the loss: Method paragraph 4 / Method paragraph 5 (end);
- Improve the explanation of the predictor and how it is influenced by the augmentations and how the augmentations are encoded: Architecture itemize 1 (predictor) / Encoding of the augmentations;
- Encoding of the augmentations (previous section 2.3.2) **moved in the appendix**: in order to clarify this point and give sufficient details to make it comprehensive, we moved it in the appendix so we had more place to elaborate and give examples;
- Justification for the choice of the invariance methods and why Barlow Twins is only applied to CIFAR10: Architecture paragraph 1 (start);
- Justification of the use of linear evaluation and training procedure: Linear evaluation paragraph 1;
- Mention the value of $\lambda$: Experimental settings paragraph 1;
- Reference to the hyperparameter study added in the appendix (A.2): Influence of the architectures paragraph 1 (end);
- Clarification on the metrics Eq. 5/6 (e.g. for negative values): Equivariance measurement paragraph 2;
- Fig. 3/4 precision on the dashed line;
- Explicit mention of fine-tuning as a perspective for future work: Conclusion and perspectives paragraph 3;
- Grammar and typos correction on the whole article (including Fig. 3/4 references).

We hope that theses improvements will answer your relevant remarks.

Best.

---

### Author Response · Authors · 2022-12-13
**Latest results of additional experiments**

Dear Reviewers,

We would like to thank you one last time for your time and your great remarks that helped us to improve our paper.

For your information, we made a first attempt at running BYOL during a full 1000 epochs on ImageNet. Results under linear evaluation are the following: Our BYOL reproduction obtains 74.03% (while the BYOL paper claim 74.3%) and BYOL + EquiMod achieves 73.22%. It is the first time that EquiMod does not improve the performance of such an invariance method, as previous experiments showed that EquiMod improves BYOL when trained only for 100 and 300 epochs. Still, note that we have only tested a single hyperparameter setup with BYOL + EquiMod 1000 epochs (while it required more iterations to find good parameters during our other experimentations with other baselines), we are currently exploring other sets of hyperparameters and may be able to find a better one by the camera-ready deadline.

Best.

---

### Decision · Program_Chairs · 2023-01-20

**Decision:**

Accept: poster

**Justification For Why Not Higher Score:**

There is still the outlier reviewer and I am not certain about it being at spotlight level, however happy to reconsider.

**Justification For Why Not Lower Score:**

N/A

**Metareview: Summary, Strengths And Weaknesses:**

The authors introduce a generic equivariance module that structures the learned latent space and show that is is directly applicable to state-of-the-art invariance models, such as SimCLR and BYOL, and that it increases performance on the CIFAR10 and ImageNet datasets.
The paper is also well written and easy to follow.
Although one reviewer scoring low there has been a very good response, in my opinion, from the authors and no response or acknowledgment thereof from the reviewer which is interpreted as a positive response in this case.
I therefore am pleased to accept this work for publication.


**Note From Pc:**

if the above contains the word "oral" or "spotlight" please see: "oral" presentation means -> notable-top-5% and "spotlight" means -> notable-top-25%. As stated in our emails, we are disassociating presentation type from AC recommendations